# Assessing the Epidemiology of Rotavirus A, B, C and H in Diarrheic Pigs of Different Ages in Northern Italy

**DOI:** 10.3390/pathogens11040467

**Published:** 2022-04-14

**Authors:** Elena Ferrari, Cristian Salogni, Vito Martella, Giovanni Loris Alborali, Alessandra Scaburri, Maria Beatrice Boniotti

**Affiliations:** 1Istituto Zooprofilattico Sperimentale della Lombardia e dell’Emilia Romagna “Bruno Ubertini”, Via Bianchi 7/9, 25124 Brescia, Italy; cristian.salogni@izsler.it (C.S.); giovanni.alborali@izsler.it (G.L.A.); alessandra.scaburri@izsler.it (A.S.); mariabeatrice.boniotti@izsler.it (M.B.B.); 2Department of Veterinary Medicine, University Aldo Moro of Bari, 70010 Valenzano, Italy; vito.martella@uniba.it

**Keywords:** rotavirus groups, pig, diarrhea, epidemiology, Italy, enteric pathogens

## Abstract

Rotaviruses are classified in 10 groups (A to J), where rotavirus A (RVA) is the major cause of diarrhea in humans and animals. With some exceptions, there is scarce information on the epidemiology of non-A rotaviruses in human and animal hosts. Currently, five species (A, B, C, E and H) have been identified in pigs. In the present study we investigated the prevalence of RVA, RVB, RVC and RVH among diarrheic pigs of different ages, in different seasons and in the presence of co-infections. Two molecular assays were developed for the detection of porcine RVA, RVB, RVC and RVH and were used to screen 962 stool specimens from suckling, weaning and fattening pigs with acute enteritis. Overall, rotaviruses were detected in a high percentage of samples (78%), with RVA being predominant (53%), followed by RVC (45%), RVB (43%) and RVH (14%). RVA was more common in the suckling (58%) and weaning cohorts (64%), while RVB, RVC and RVH were also frequently detected in fattening pigs. Only RVA and RVB infections followed a seasonal trend and exhibited age-related differences. Rotavirus infections were frequently present in combination with other pathogens. The present study depicts a portrait of rich rotavirus diversity in porcine herds, identifying seasonal and age-related patterns of circulation of the different rotavirus species in the surveyed areas.

## 1. Introduction

Rotaviruses (RVs), members of the *Reoviridae* family, are non-enveloped viruses with a trilaminar capsid of about 70 nm in diameter enclosing 11 segments of double-stranded RNA, and on microscopy observation they typically appear as wheel-shaped particles. Since their discovery in the early 1970s, it was noted that, based on VP6-mediated antigenic reactivity and on the pattern of migration of the 11 genome segments in gel electrophoresis, there were different RV groups. Based on VP6 gene sequence [1] and, later on, on full-genome sequencing [2,3], 10 different RV groups or species (A to J) are now recognized, variously distributed in the human and animal hosts. Among these, RVA to RVC, RVE, RVH, RVI and RVJ are known to infect mammals, and RVA is the most common species described in most, if not all, mammalian hosts [2]. RVAs are major human enteric pathogens, and vaccines are now used globally to prevent acute gastroenteritis in children [4]. Several examples of animal-to-human transmission of RVAs have been reported [5,6], and a major epidemic equine-derived RVA strain has emerged globally in humans after 2013 as a reminder of RV zoonotic potential [7].

Rotaviruses have been also described in piglets worldwide [8]. In porcine farms, RV-associated enteric diseases represent one of the main causes of morbidity and mortality, resulting in relevant economic losses. Currently, five groups (RVA, RVB, RVC, RVE and RVH) have been identified in fecal specimens from pigs of various ages affected by diarrhea [9,10,11,12,13,14]. Rotavirus A (RVA) is the most common and is characterized in term of epidemiology, genetic variability and pathogenicity. RVA infections predominate in nursing piglets, but recent studies also described a high frequency in post-weaning pigs and were found with or without association to diarrhea [15,16,17]. RVC has been reported worldwide in diarrheic outbreaks in nursing, weaning and post-weaning pigs [13,18,19,20,21,22]. RVB infections have been described in pigs with enteric signs from South and North America (33%) and from Japan (25.9%) [9,23,24]. Epidemiological data on RVB in European countries are scarce. Low rates of RVB infections have been reported in pigs with enteritis in the Czech Republic (0.6%), Germany (1.6%) and in Spain (9.3%) [11,13,25], while a higher prevalence has been observed in adult pigs [9]. RVB is a highly variable group, of which nine different VP6 genotypes were described in pigs [26]. RVE was reported three decades ago in a porcine herd in the UK [14], but, after its first identification, it has never been identified elsewhere. Moreover, sequence data of RVE have not been generated, and this historical RV species (group) remains to be confirmed. RVH infections have been sporadically described in adult pigs in Japan, USA (15%) and Brazil (18%) [10,27,28]. Data about RVH epidemiology are not available in Europe.

Despite enteric diseases being more often associated with cold seasons, porcine RV infection did not show seasonality, but, rather, spatio-temporal oscillations, suggesting their possible dependence on other factors. In addition, the epidemiology of RV enteritis is complicated by the presence of mixed infections, with other enteric pathogens likely increasing the severity of the disease. In particular, RV co-infections with PEDV, *Clostridium difficile*, *perfrigens* and *E. coli* have been described in several studies [15,25,29].

Overall, the information on RV epidemiology in pigs is fragmentary, and relevant information is missing for non-A RVs, as the available diagnostic tools primarily target RVA. In order to investigate the epidemiological patterns of porcine RVs, we developed two quantitative multiplex assays based on reverse-transcription (RT) polymerase chain reaction (PCR) for the simultaneous detection of RVA and RVB and of RVC and RVH. The quantitative RT-PCR (RT-qPCR) assays were used to screen the stools and/or the intestinal content of pigs from episodes and outbreaks of enteritis that were monitored during passive surveillance in areas of the north of Italy.

## 2. Results

### 2.1. Development of RV Multiplex RT-qPCR Assays

RVA, RVB, RVC and RVH individual assays were designed, optimized and combined into two multiplex RT-qPCR: RVA-RVB and RVC-RVH. The in silico analysis of oligonucleotides that was designed in our study, conducted on RV VP6 porcine sequences deposited in the NCBI GenBank, showed a complete match to the RVA I5 genotype; RVB I11, I12 and I13 genotypes; RVC I1, I4, I5, I6, I7, I10, I11 and I13 genotypes and RVH I1 genotype. The efficiency of the assays was evaluated by the construction of four individual standard curves for RVA, B, C and H. The regression analyses showed that their efficiency was well matched (R^2^ > 0.99) and all were above 95% (Table 1). The sensitivity, tested using limiting dilutions of positive control samples, displayed detection limits equivalent to 10^2^ copies per reaction for all the targets. On the basis of the calculated LOD, a Cq cut-off of 37 was set. The specificity of both the assays was confirmed by the lack of amplification by RT-qPCR of a panel of 15 isolates of porcine enteric pathogens different from RV (Appendix A). To assess the absence of contaminants in the extraction and the reliability of the RT-qPCR assays, a negative extraction control, no template control (NTC), the internal controls (IC) and a positive control were analyzed in each run. Both the multiplex assays successfully amplified the positive and internal controls, while no amplification curves were observed with the NTC and negative extraction control.

### 2.2. RV Group Distribution among the Age Classes

Between November 2016 and December 2019, 78% (750/962) of the submitted specimens were positive for RVs. RVA was detected in 56.8%, RVB in 43%, RVC in 45.3% and RVH in 14.1% of the samples (Appendix A). The majority of RV-positive cases presented a combination of RV species (60.7%, 455/750). Among single RV infections (39.3%, 295/750), RVA was predominant (17.2%), followed by RVC (6.9%), RVB (5.7%) and RVH (0.9%). In order to estimate the distribution of RV by age category, the samples were divided into suckling, weaning and fattening classes (Figure 1). In the suckling class, RVA was the most common (57.9%, 184/318), mostly as single infection (34.3%). In the weaning class, RVA and RVC were more prevalent (64.4%, and 60.3%, respectively), frequently in mixed infections with other RV species. Mixed infections accounted for 53.1% of RVA- and 52.7% of RVC-positive samples. RVA, RVB and RVC were equally distributed across the fattening pigs (51.4%, 52.3% and 55.1%, respectively) and were mostly detected in mixed infections. RVH-positive cases increase with age, ranging from 3.1% in suckling piglets to 22% in fattening pigs. Interestingly, the majority of RVH infections (93.4%, 127/136) were in combination with the other RV species. As shown in Table 2, RV group distribution seems to be influenced by the age class. The prevalence of RV infection was significantly higher in the weaning class (84.5%) than in the suckling class (72%, *p* = 0.0005). RVA was predominant in the weaning class (64.4%), and the difference was significant with respect to the fattening class (51.4%, *p* = 0.0012). Moreover, the prevalence of RVB and RVH was higher in the fattening and weaning classes than in the suckling cohort (*p* < 0.0001 for both the comparisons). RVC was more common in weaning (60.3%) and fattening pigs (55.1%) than in suckling pigs (21.7%, *p* < 0.0001 for both). Considering the distribution of RV as single or mixed infections, the prevalence of RVA in suckling piglets was significantly higher than that observed in the other classes (*p* < 0.0001). In contrast, the numbers of RVB, RVC and RVH mixed infections were higher in the weaning and fattening cohorts with respect to the suckling cohort (*p* < 0.0001).

### 2.3. Seasonal Distribution of RV among Age Classes

Due to the small number of analyzed samples for each month of the study period (2016–2019), all RV cases were aggregated into seasons. In the suckling class, a peak of RV infections was registered during the spring, and the lowest number of cases was registered during the summer (Figure 2). Among the RV groups, RVA showed the most pronounced seasonality, with a peak activity in spring (*p* < 0.0001, Appendix A). While in the suckling cohort the trend of RV infections matched the trend of clinical cases, in the weaning and fattening classes the pattern of RV infections was not proportional to the total number of enteric cases (Figure 2). Notably, in weaning piglets, only RVB showed a seasonal distribution, with a peak of infections during the spring and autumn (Figure 2, Appendix A). The RV infection rate among the fattening pigs was constant from spring to autumn. Considering the RV group distribution, RVA and RVB showed a lower rate during winter with respect to the remaining part of the study period. In conclusion, our data showed significant seasonal fluctuations for RVA and RVB in all age classes, while RVC and RVH were more scattered throughout the entire study period.

### 2.4. Co-Infections of RV with Other Pathogens

According to the age class (young and adults), different viral and bacterial enteric pathogens were investigated in combination with RV infection. A sub-set of 110 and 41 suckling and weaning piglets, respectively, were analyzed for *E. coli*, *C.difficile*, *C.perfringens* and PEDV, while 163 fattening pigs were analyzed for *E. coli*, PEDV and pathogenic bacteria causing gastroenteritis in adult pigs (*Brachyspira hyodysenteriae*, *Brachyspira pilosicoli and Lawsonia Intracellularis*). The number of positive samples for RV co-infections and single infections for each age class is summarized in Table 3. Briefly, RV single infections were detected in 5.5% (6/110) of suckling, 14.6% (6/41) of weaning and 1.4% (2/163) of fattening pigs. Conversely, RV mixed infections were observed in 68.2% (75/110) of the suckling class, in 63.4% (26/41) of the weaning and 85.3% (139/163) of the fattening class. For all age categories, *E. coli* was the agent most associated with RV infections, but not all the E.coli strains were characterized as pathogenic. Its presence was detected in 60% (66/110) of suckling, 41.5% (17/41) of weaning and 81% (132/163) of fattening pigs.

## 3. Discussion

Rotaviruses are highly diffused among pigs with enteric disease and have been associated with increased mortality, growth impairment and economic losses [13,20]. Data on the distribution of RV infections in European countries, chiefly for RVB, RVC and RVH, are scarce or not available at all.

A major limit for the generation of epidemiological data is represented by the availability of reliable diagnostic tools. While several direct diagnostic assays are available for the detection of RVA antigen and RNA, this does not apply to non-A RV species. Moreover, the high genetic variability of RVs and, for RVB and RVH, the limited number of genomic sequences [23,30,31,32] may challenge the development of primers and probes. To fill these diagnostic gaps, we developed two multiplex RT-qPCR assays for the specific detection of porcine RVA, RVB, RVC and RVH. However, due to the high sequence variability of RVA, RVB and RVH, not all the porcine genotypes described in the public database can be detected by the RT-qPCR assays. These molecular tests were used to screen a large collection of samples collected during 2016–2019 in Italian herds from pigs of different ages with signs of enteritis. In general, we identified RV infections in 78% of pigs affected by enteritis, with the highest number of cases in weaning pigs (85%). Indeed, during the weaning phase, piglets are more susceptible to infections due to a major exposure to stress factors (dietary changes, removal from the sow and changes in the environment) that may negatively affect the response of the immune system [33,34]. Despite RVA infections being more common in the weaning pigs, RVA single infections were mostly detected in suckling pigs (34%), suggesting a primary role of RVA infection in this age class. Unlike previous observations that RVC infection is more common in suckling piglets [9,18,20,22], in our investigation we observed the highest prevalence in weaning and fattening pigs (60% and 55%, respectively). Low rates of RVB infections were detected in pigs from Germany (1.6%), Czech Republic (0.6%) and Spain (9.3%) [11,13,25]. Interestingly, there was an overall higher prevalence of RVB cases (43%) in our study than in other European investigations, suggesting an increase of RVB prevalence over the period and in the locations surveyed in this study. However, it cannot be ruled out that this contrasting data could be due to an underestimation of the real diffusion of RVB. It must be considered that RVB is a highly variable group, of which nine different VP6 genotypes were described in pigs; therefore, its detection rate could be affected by primer design [26].

This would require large, structured investigations and, eventually, the use of sensitive and specific diagnostic tools. In addition, in line with other studies, in our investigation we identified most RVB infections in the older age classes, i.e., in weaning and fattening pigs [9,24]. The presence of RVB mixed infections was also significantly higher in the weaning and fattening classes, suggesting that RVB could act as a secondary diarrhea agent in older pigs. RVH was detected at a lower rate (14%) than other RV species, and it was more common in older animals (weaning and fattening classes). In agreement with previous works, RVH infections were mostly detected in combination with other RV groups, strengthening the hypothesis of a role as an opportunistic agent in swine enteric diseases [9,24].

Interestingly, in our investigations a seasonal influence was observed on the prevalence of RVA and RVB infections in pigs affected by enteritis, with some difference depending on the age class. Notably, RVA reached a peak of activity during the spring in suckling pigs and two peaks (during spring and autumn) in fattening pigs. Similarly, RVB activity peaked during spring and autumn in the weaning cohort and steadily increased between spring and autumn in fattening pigs. By contrast, RVC and RVH did not seem to be subjected to seasonal variations. Interestingly, in suckling pigs, RV infections matched the trend of cases of enteritis, suggesting an association between the number of enteritis and RV cases. In contrast, in weaning and fattening pigs, RV activity did not match the cases of enteritis reported to our diagnostic laboratories, suggesting a minor role of RV in the etiology of enteritis in older animals. Previous works described oscillations of RVA and RVC during the years without any seasonal peaks [16,19]. Similarly, a multilevel model study conducted from 2009 to 2011 that considered influencing factors such as geographical location and age reported fluctuations of RVA and RVB infections [35]. Probably, other factors, such as the presence of co-infections may concur with a different susceptibility of RV infections over the seasons and years.

Since diarrhea may be caused or exacerbated by multiple enteric pathogens, we also focused on the frequency of co-infections. Previous studies reported that enteritis in young piglets is usually related to a single pathogen [8]. In contrast, in the present study, we reported a higher percentage of mixed infections in the young classes (89% of suckling pigs and 77.5% of weaning pigs). Among co-infections, RV was highly present in the fattening class (85%) and in the suckling and weaning classes (68% of suckling and 63% of weaning). These data suggest that, especially in adults, enteric disease could be the result of a combination of pathogens. In general, *E. coli* was the agent that was most associated to RV in all age groups, however, the *E. coli* pathotype was fully characterized for only a small number of samples, hindering the understanding of its role in infections.

RVs seem to play a primary role in the etiology of diarrhea, mostly in the weaning class, where it was detected as the higher percentage of RV single infections (14.6% vs. 5.5% of suckling pigs and 1.2% of fattening pigs). However, it is worth considering that other viral and bacterial enteric organisms (i.e., astroviruses, picornaviruses, caliciviruses, parvoviruses and so on) not included in the present analyses could eventually interfere and puzzle the observed scenario. Despite this, the study underlines the variability and complexity of the epidemiological picture. It does not allow the specific role of RV in the etiology of disease to be addressed. A limit of our investigation, based on passive surveillance, is surely the lack of a control group of healthy animals. Studies including a case-control group should be conducted to more properly analyze the association between enteritis, RV and co-infections.

In summary, the present study reported a high prevalence of different RV groups in Italian pig herds affected by enteric disease. Differences in prevalence among age classes and the various RV groups and seasonal peaks of activities for RVAs and RVBs were observed. The high detection of RVB, RVC and RVH suggests that non-A RV species may also play an important role in the etiology of swine enteritis. Implementing the diagnostic algorithms of swine enteritis with non-A RVs could help to better understand their role and devise, eventually, strategies suitable for their control in the herds.

## 4. Materials and Methods

### 4.1. Samples

Between November 2016 and December 2019, feces and small intestines from pigs of various ages were submitted by veterinarians or farmers to determine the causative agent(s) of enteritis cases from porcine herds in Italy. Screening for RVs was conducted on 962 specimens. The samples were categorized by age class in suckling piglets (<8 kg body weight), weaning piglets (8–25 kg body weight) and fattening pigs (>25 kg body weight). The collected samples were diluted 1:10 (*w*/*v*) in minimum essential medium (MEM) and homogenized by vortexing. Suspensions were clarified by centrifugation for 10 min at 4.000× *g* to eliminate debris. Next, 200 μL of the supernatant was used for RNA extraction with a commercial kit (NucleoMag^®^ Vet kit, Macherey-Nagel, Düren, Germany) following the manufacturer’s instructions.

### 4.2. Quantitative RT-qPCR for Detection of RVs

RT-qPCR primers and hydrolysis probes were designed based on porcine RVA, RVB, RVC and RVH VP6 sequences that are available in the NCBI GeneBank database. RVA, RVB and RVH but not RVC oligonucleotides, contain degenerated nucleotides to allow the detection of a large number of RV sequences. Two multiplex RT-qPCR assays were developed for the simultaneous detection of RVA/RVB and RVC/RVH using the primers and probes reported in Table 4. RT-qPCR assays were performed in a “one-step” format using the commercial QuantiFast Pathogen RT-qPCR kit (Qiagen, Hilden, Germany). An exogenous internal control RNA (IC) and a specific detection assay, provided by the manufacturer, were added to each RT-qPCR reaction to verify the absence of inhibitors. Briefly, 6 µL of extracted RNA were added to 2.5 µM of each RVA, RVB primer (for RVA-RVB assay) or to 2.5 µM RVC primers and 3.75 µM RVH primers (for RVC-RVH assay) to a final volume of 10 µL. Then each mix was subjected to denaturation at 95 °C for 5 min. Next, 5 µL of the denaturation mix was added to 20 µL of the RT-reaction mix, which was prepared following the manufacturer’s instructions. The RNA was reverse transcribed at 50 °C for 20 min, followed by one cycle of Taq polymerase activation at 95 °C for 5 min. Amplification consisted of 45 cycles at 95 °C for 15 s and 60 °C for 30 s. The RT-qPCR assays were performed with the CFX96 Touch Real-Time PCR Detection System (Bio-Rad Laboratories, Hercules, California, USA) and the data were analyzed with the Software Bio-Rad CFX Manager 3.1 using the single threshold method for the Cq determination. An experiment was accepted when the Cq of the “no template control” (NTC) and negative control of extraction were not determinable or >40, the internal control (IC) of samples was <34 and the positive control ranged between 28 to 33 Cq.

### 4.3. RV RT-qPCR Performances

The limit of detection (LOD) of the multiplex RT-qPCR assays was determined by testing six replicates of 10-fold serial dilutions of RVA-, RVB-, RVC- and RVH-positive samples at a known genomic copy titer, starting with an initial concentration of 10^4^ to a final concentration of 10 gene copies/reaction. The LOD was calculated as the lowest RNA concentration that could be detected in all six replicates. The specificity of the two assays was determined by a RT-qPCR analysis of a panel of 15 isolates of porcine viruses and bacteria that are causative of enteritis in swine. VP6 RVA-, RVB-, RVC- and RVH-specific gene fragments were cloned, in vitro transcribed and quantified by a spectrophotometer (Infinite^®^ 200 NanoQuant, Tecan, Männedorf, Switzerland). Serial dilutions of such transcripts (10^7^ to 10^3^ genomic copies for reaction) were run on the RVA-B and RVC-H RT-qPCR assays to generate a standard curve and determine the efficiency of the reactions and the linear correlation index (R^2^).

### 4.4. Bacterial and Viral Agent Detection

Subsets of fecal and intestinal samples was also tested for porcine epidemic diarrhea virus (PEDV), *Escherichia coli*, *Clostridium perfringens*, *Clostridium difficile*, *Lawsonia intracellularis*, *Brachyspira pilosicoli* and *Brachyspira hyodisenteriae*. PEDV was detected by RT-qPCR as described previously [36]. *E. coli* and *C. perfringens* were detected by direct culture that was incubated in anaerobic atmosphere at 37 °C for 24 h (5% sheep blood Columbia agar). Identification took place by standard phenotypic characterization, including the analytical profile index system. *C. difficile* was detected by direct culture, after a treatment of the caecum content with 95% ethanol 1:1 (*w*/*v*) for 30 min using 5% sheep blood Columbia agar that was incubated in an anaerobic condition for 48h. For the detection of *Lawsonia intracellularis*, fecal and intestinal samples were diluted in PBS1X 1:10 (*w*/*v*). Frozen intestine was homogenized by mortar and pestle, while stools were homogenized by vortexing. DNA was extracted by a DNeasy^®^ Blood and Tissue Kit (Qiagen, Hilden, Germany) according to manufacturer’s instructions. The extracted DNA was analyzed by a nested PCR assay as described in a previously published method [37]. *Brachyspira* ssp. was detected by direct culture using selective media (Brachyspira agar) addicted with specific antibiotics (rifampicin, vancomycin and colistin) in an anaerobic atmosphere for 5 days. Bacterial colonies were then submitted to a multiplex real-time PCR method for simultaneous identification on the basis of the previous work [38].

### 4.5. Statistical Analysis

The prevalence of RV infection and the 95% confidence interval were calculated for each age class using a binomial model or the Clopper–Pearson method when the size of the samples was less than 10. The associations between RV and age classes were analyzed by a χ2 test or Fisher’s exact test when the size of the samples was less than 5. RV prevalence was also calculated for each season of the entire period of the study, with winter from December to February, spring from March to May, summer from June to August and autumn from September to November. The association between RV cases and season was calculated by a χ^2^ test. All statistical analyses were performed assuming a 5% significance level using R software, 3.6.1 version [39].

## Figures and Tables

**Figure 1 pathogens-11-00467-f001:**
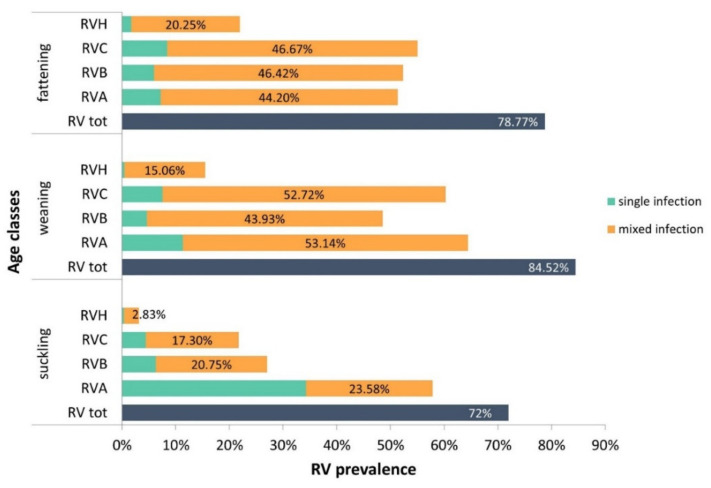
Distribution of overall RV, RVA, RVB, RVC and RVH across the suckling (n = 318), weaning (n = 239) and fattening class (n = 405) as single and mixed infections.

**Figure 2 pathogens-11-00467-f002:**
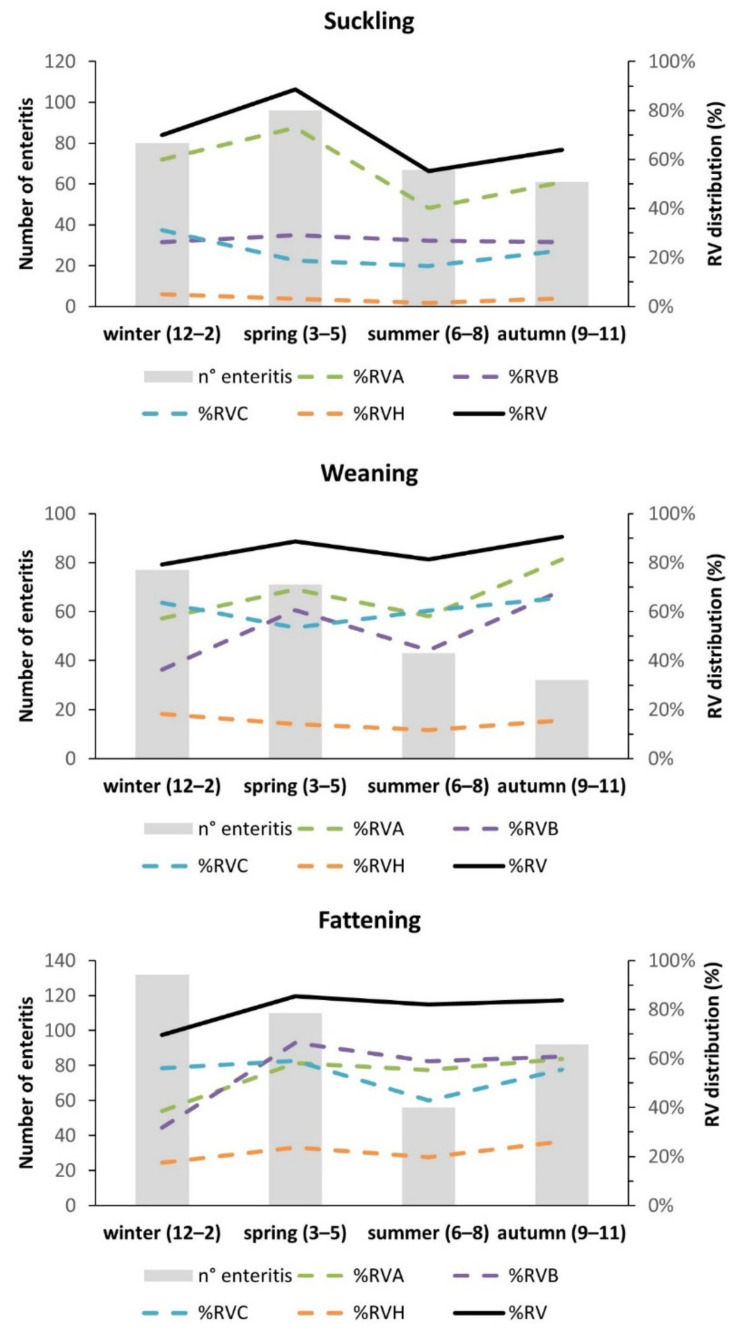
Seasonality pattern of RV groups among suckling, weaning and fattening pigs. Positive RV cases (indicated by bars) tested from December 2016 to November 2019 were grouped into seasons. Winter ranges from December to February (month 12–2), spring from March to May (month 3–5), summer from June to August (month 6–8) and autumn from September to November (month 9–11). The prevalence of RV is shown by lines.

**Table 1 pathogens-11-00467-t001:** Performance parameters of the RVA-RVB and RVC-RVH multiplex RT-qPCR assays.

	RVA-RVB Assay		RVC-RVH Assay	
Test	RVA	RVB	RVC	RVH
Slope ^§^	−3.3896 ± 0.0495	−3.428 ± 0.027	−3.309 ± 0.019	−3.263 ± 0.045
Calibration curve (R^2^) ^§^	0.9996 ± 0.0001	0.999 ± 0.0002	0.999 ± 0.001	0.998 ± 0.001
PCR efficiency (%) ^§^	97.093 ± 1.943	95.770 ± 1.035	100.6 ± 0.839	102.6 ± 1.950
Specificity (%)	100	100	100	100
LOD (genomic copies) ^‡^	100	100	100	100

^§^, mean of three independent replicate standard curves; R^2^, linear correlation index; LOD, limit of detection; ^‡^, mean of two independent replicates.

**Table 2 pathogens-11-00467-t002:** Statistical analysis of RV group distribution among suckling (S), weaning (W) and fattening (F) pigs.

RV Group	χ2	*p*-Value	S vs. W	S vs. F	W vs. F
**RV (all groups)**	12.68	0.002 *	0.0005 *	0.035 *	0.070
**RVA total infections**	10.71	0.005 *	0.259	0.190	0.001 *
RVA single infections	99.85	<0.0001 *	<0.0001 *	<0.0001 *	0.070
RVA mixed infections	55.99	<0.0001 *	<0.0001 *	<0.0001 *	0.028
**RVB total infections**	50.44	<0.0001 *	<0.0001 *	<0.0001 *	0.619
RVB single infections	0.78	0.679	0.390	0.839	0.474
RVB mixed infections	56.11	<0.0001 *	<0.0001 *	<0.0001 *	0.540
**RVC total infections**	108.61	<0.0001 *	<0.0001 *	<0.0001 *	0.404
RVC single infections	4.67	0.097	0.136	0.323	0.461
RVC mixed infections	48.12	<0.0001 *	<0.0001 *	<0.0001 *	0.100
**RVH total infections**	52.51	<0.0001 *	<0.0001 *	<0.0001 *	0.113
RVH single infections ^†^	N.D	N.D	1	0.085	0.269
RVH mixed infections	48.18	<0.0001 *	<0.0001 *	<0.0001 *	0.100

* statistically significant results (*p* < 0.05); N.D, not determined; ^†^, For samples with n < 5, the Fisher’s exact test was calculated.

**Table 3 pathogens-11-00467-t003:** Prevalence of enteric pathogens in subsamples of suckling, weaning and fattening pigs.

	Suckling (n = 110)		Weaning (n = 41)		Fattening (n = 163)	
	n	%	n	%	n	%
**Single infections**	**10**	**9.1%**	**9**	**22.0%**	**7**	**4.3%**
RV	6	5.5%	6	14.6%	2	1.2%
E.coli	2	1.8%	2	4.9%	5	3.1%
PEDV	1	0.9%	1	2.4%	0	0.0%
C.difficile	1	0.9%	0	0.0%	-	-
C.perfrigens	0	0.0%	0	0.0%	-	-
B.pilosicoli	-	-	-	-	0	0.0%
B.hyodisenteriae	-	-	-	-	0	0.0%
L.intracellularis		-	-	-	0	0.0%
**RV mixed infections**	**75**	**68.2%**	**26**	**63.4%**	**139**	**85.3%**
RV+E.coli	66	60.0%	17	41.5%	132	81.0%
RV+PEDV	6	5.5%	9	22.0%	26	16.0%
RV+C.difficile	28	25.5%	4	9.8%	-	-
RV+C.perfrigens	32	29.1%	6	14.6%	-	-
RV+B.pilosicoli	-	-	-	-	7	4.3%
RV+B.hyodisenteriae	-	-	-	-	27	16.6%
RV+L. intracellularis	-	-	-	-	32	19.6%
**non RV co-infections**	**23**	**20.9%**	**5**	**12.2%**	**17**	**10.4%**
**negatives**	**2**	**1.8%**	**1**	**2.4%**	**0**	**0.0%**

**Table 4 pathogens-11-00467-t004:** Sequence of primers and hydrolysis probes used in RT-qPCR assays with relative concentrations.

RT-qPCR Assays	5′–3′ Sequence	Conc. (nM)
RVA-RVB assay	RVA_VP6_for CACCTTCAAGAGARGATAAYTTRCAA	500
	RVA_VP6_rev TCGGATACCAGGTRKTTAGCCT	500
	RVA_VP6_probe FAM-TCCATTAGAAGCATGTTGAT-MGB	200
	RVB_VP6_for TRTGGKGWCARAARATAGCRAT	500
	RVB_VP6_rev ACCTYTCGAAGCACTYCCWTT	500
	RVB_VP6_probe VIC-TGATCCGGCGTCRGCT-MGB	100
RVC-RVH assay	RVC_VP6_for TGTAGCATGATTCACGAATGGGT	500
	RVC_VP6_rev ACATTTCATCCTCCTGGGGATC	500
	RVC_VP6_probe CY5-GCGTAGGGGCAAATGCGCATGA-BHQ2	100
	RVH_VP6_for CCACCACAATTMGTTCAYTGGTC	750
	RVH_VP6_rev TCCCAGTGCGTGACCAGAT	750
	RVH_VP6_probe FAM-GCATGTTTAATTGCAGCYTATTC-MGB	200

## Data Availability

Not applicable.

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
