# Peer review of "Assessing the Epidemiology of Rotavirus A, B, C and H in Diarrheic Pigs of Different Ages in Northern Italy"

_pathogens, 2022, doi:10.3390/pathogens11040467_

Round 1
Reviewer 1 Report
This manuscript describes trends of RVs detections by age and seasons via passive surveillance using multiple fecal samples collected in North Italy. The data presented in this study are interesting and useful for control RV infections in pig herds. Therefore, the information obtained in this study is valuable in publication in this journal. However, this manuscript has one major problems.
Major revision:
RVA, RVB, RVC, and RVH have been detected all multiple genotypes in pigs.
Are the primers used in this study designed to can detect all kinds of genotypes in individual RVs.
The design of primers is enough affected on detection rates of each RVs in pig samples.
Therefore, the authors should test and describe in detail genotype-dependency of primers used in all RV species.
Minor revision:
Line 54-56 and Line 192-193
Especially, porcine RVBs have multiple different genotypes.
Low prevalence in those countries may influenced by design of primers.
Please reconsider these sentences and citations.
Reviewer 2 Report
The authors investigated the prevalence of rotavirus A, B, C, and H in diarrheic pigs by age category (suckling, weaning, and fattening classes), in different seasons, and in presence of co-infections. The authors developed two quantitative multiplex assays (RT-qPCR) for simultaneous detection of rotavirus A and B, and rotavirus C and H, respectively.
Minor revision:
Page 2, line 74: change “(qRT-PCR)” to “(RT-qPCR)”.
Page 2, line 85 and Supplementary Table S1: change “Ct” to “Cq”.
Page 3, line 95: change “indipendente replicate” to “independent replicate”.
Page 3, line 112: change “table 2” to “Table 2”.
Page 7, line 161: change “table 3” to “Table 3”.
Page 10, line 289: change “RT-PCR assays” to “RT-qPCR assays”.
Page 10, line 308: change “for 30´” to “for 30 minutes”.
Page 11, line 334: change “Aknowledgements” to “Acknowledgements”.
There are some other typos, also including punctuation, to be corrected in the manuscript.
Author Response
Reply to reviewers’2 comments:
- Page 2, line 74: change “(qRT-PCR)” to “(RT-qPCR)”: we changed the word as suggested (line 75, revised manuscript)
- Page 2, line 85 and Supplementary Table S1: change “Ct” to “Cq”: we replaced “Ct” to “Cq” in line 89 of the revised manuscript and in Supplementary table 1 revised
- Page 3, line 95: change “indipendente replicate” to “independent replicate”: we changed the sentence as suggested (line 99, revised manuscript)
- Page 3, line 112: change “table 2” to “Table 2”: we change lower letter of table to upper letter (line 116, revised manuscript).
- Page 7, line 161: change “table 3” to “Table 3”: we changed the word as suggested (line 167, revised manuscript)
- Page 10, line 289: change “RT-PCR assays” to “RT-qPCR assays”: we re-typed “RT-PCR assays” to “RT-qPCR assays (line 275 and 282, revised manuscript)
- Page 10, line 308: change “for 30´” to “for 30 minutes”: we changed the typing of minutes as suggested (line 320, revised manuscript)
- Page 11, line 334: change “Aknowledgements” to “Acknowledgements”: we corrected Aknowledgements” into “Acknowledgements (line 346, revised manuscript)
- There are some other typos, also including punctuation, to be corrected in the manuscript. In this regard we correct any punctation mistake
Round 2
Reviewer 1 Report
The revised manuscript has been improved corresponding to reviewer's comments. The reviewer hopes that the authors will analyse and discuss trend of genotypes in individual RVs in further study following to this study.